# Internationalisation of Teaching and Learning through Blended Mobility: Potentials of Joint International Blended Courses and Challenges in Their Implementation

**René Perfölz [1] and Asun López-Varela [2,*]**

[1] Services for Teaching and Learning, Center for Digital Systems, University Library, Freie Universität Berlin, 14195 Berlin, Germany

[2] Department of English Studies, Faculty of Languages, Universidad Complutense de Madrid, 28040 Madrid, Spain

* Correspondence: alopezva@ucm.es

**Abstract:** Blended Mobility formats such as joint international blended courses have the potential to enable more students at universities and other HEIs to gain international experiences in the course of their studies. They enhance transnational cooperation in the European Higher Education Area by building bridges at the crossroads of education, research, innovation, serving society and economy. In this article, the authors reflect on their experiences in the conception, planning, organisation and implementation of a joint international blended course between Freie Universität Berlin and Universidad Complutense de Madrid in the field of sustainable development in the summer semester of 2022. The course was offered within the framework of the Erasmus+ KA3 project "Online Pedagogical Resources for European Universities" (OpenU project).

**Keywords:** blended mobility; international learning; joint international blended courses; sustainable green and digital transitions; OpenU

## 1. Introduction

The Erasmus+ KA3 project "Online Pedagogical Resources for European Universities" (Acronym OpenU, Ref. 606692-EPP-I-2018-2-FR-EPPKA3-PI-POLICY) is a consortium of European universities seeking to support international blended learning, mobility and networking in Higher Education Institutions HEIs in Europe. It, therefore, addresses two fundamental aspects that can contribute to strengthen long-term strategic and structural cooperation between European HEIs: innovative solutions for the internationalisation of educational practices, strengthening university alliances and contributing to the transferability of innovative models, and joint agendas for incorporating sustainable development.

Within the OpenU consortium, Freie Universität Berlin (FUB) and Universidad Complutense de Madrid (UCM) have developed a joint experimentation project seeking to expand the internationalisation of teaching practices in both universities and to fill a methodological gap in the inclusion of sustainable development within the internalisation of the teaching agenda of Higher Education. The following lines present the challenges found in the implementation of the project as well as its achievements.

The paper is divided in various sections that present internationalisation policies in FUB and UCM, the context of sustainable development in Europe, the methodology used in the implementation of the international joint blended course, the results achieved, their possible implications for transference to other institutions, as well as some final recommendations and conclusions.

### 1.1. Internationalisation

As international network universities, FUB and UCM offer all students the opportunity to prepare for a life in a globalised world and workplace, and a chance to experience the

value of transnational cooperation and intercultural encounters. This is mainly reached through the continuing internationalisation of the formal curriculum as well as informal kinds of learning. While FUB and UCM include programmes that involve physical mobility, opportunities are extending to meet the needs of a diverse student population through innovative, participatory, digital scenarios for teaching and study, including virtual and blended mobility [1]. In this regard, both FUB and UCM have a strong strategic interest in further developing digital learning and teaching formats within the framework of the Una Europa alliance in order to offer partners new exchange opportunities and, thus, create new opportunities for internationalisation. The research conducted is intended to make a strategic contribution to the "European Universities" initiative and the internationalisation strategies of both universities.

A large proportion of students are interested in gaining international experience as part of their studies; however, for a variety of reasons, they do not always have the opportunity to participate in long-term physical exchange programmes. Blended Mobility (BM) has the potential to offer all students the opportunity to gain international experience in a flexible, inclusive, and innovative way: it combines joint online teaching and learning phases with periods of short-term physical mobility at one or both partner universities. Through the online exchange with international students and teachers, students who, for instance, do not have time to participate in a physical exchange programme, can still gain international experience. At the same time, they can participate in a physical short-term mobility at their international partner university, which increases the motivation for long-term participation and cooperation. The face-to-face exchange again can be made more sustainable through digital preparation phases and follow-up communication. Brief: BM provides easier, low-threshold access to international encounters through digital exchange and, at the same time, creates incentives for long-term student motivation and participation through physical short-term mobilities. However, the effort required to implement BM formats is not insignificant and there are numerous challenges that need to be addressed at an organisational, technical and policy level, to foster their development and implementation at HEIs.

The experimentation carried out between FUB and UCM aimed first and foremost at identifying challenges for teachers and teaching support services in designing, planning, organising and teaching an international joint blended course. These courses are usually taught by two or more teachers from different partner universities, alternating their teaching or simultaneously teaching in tandem, and involving two or more student groups from the respective universities. Another goal was to determine the motivation of students to participate in a blended international course, as well as the course's potential in enabling students to gain international experiences. Furthermore, the experimentation was to help determine which requirements need to be placed on the joint European digital hub, which is to be developed within the framework of the OpenU project: BLOOM. Possible aspects include joint teaching online tools, access to shared teaching materials, online communication, and collaboration between student groups from different international universities or tools for flexible online communication exchanges. Finally, the experimentation also sought to explore which policies would need to be adapted at HEI and European level in order to foster the development of joint international blended courses. Our findings are of particular value for the development of future blended mobility formats, not only in the context of the Una Europa Alliance but also within the European Higher Education Area as a whole.

*1.2. Sustainable Development in the EU Context*

In the last decade, Higher Education Institutions were looking for ways of enhancing the connections between curricular activities and sustainable development. On 22 November 2016, the EU presented its response to the United Nations 2030 Agenda [2–4] and adopted a set of priorities for sustainable development [5,6]. Among the decisions adopted, and in line within SDG 4 [7], target 4.7 [8], educational institutions should make sure that:

All learners should have acquired through education the knowledge and skills needed to promote sustainable development, including, among others, sustainable habits and lifestyles, knowledge with regard to the defence of human rights, gender equality, the promotion of a culture of peace and non-violence, an appreciation of cultural diversity and for culture's contribution to sustainable development [8].

However, these reports and indicators recognize that there is an urgent need to strengthen the introduction of sustainability in education across world regions [3,4].

The experimentation presented in this paper defends that the contribution of cultural products to sustainable development needs to bridge STEM and STEAM disciplines (Science, Technology, Engineering, Arts, Mathematics). The main reasons behind this claim are related to the fact that many of the soft skills required in sustainable development come from areas within the Social Sciences and Humanities. A STEAM approach can help students access science concepts from different vantage points, promoting creative thinking, and enhancing commitment and understanding.

Many of the 17 Sustainable Development Goals (SDG) within the United Nations framework cannot be measured only in terms of economic value. Instead, they require opportunities and ideas to be transformed into value for others. The created value can be financial, cultural, or social. This way of measuring value in terms of quality is characteristic of the Social Sciences and Humanities (SSH). The introduction of soft skills [9–11] and responsibility in research and innovation addresses wider social impact challenges on sustainable development [12], marked also by the MoRRI indicators [13], aligned with issues related to social value [14] and Responsible Research and Innovation RRI [15].

Additionally, the UN resolution on Creative Economy for Sustainable Development highlights the sector as an important tool for the attainment of the Sustainable Development Goals (UNCTAD) [16]. According to recent forecasts, the creative economy is one of the most rapidly growing sectors of the world economy; in particular, concerning income generation and job creation [17]. It will represent around 10 percent of global GDP in the years to come [18,19]. Creative economy also generates non-monetary value that contributes significantly to achieving people-centric, inclusive and sustainable development [20]. As noted, this value comes from areas related to SSH education and requires a systemic approach that connects individuals, in this case students in Higher Education Institutions, with endeavours in art, design, culture and heritage, within sustainable development.

## 2. Methodology: Experimentation Description

The aim of the FUB-UCM experimentation was firstly to gather experience on the practical implementation of joint international blended courses. The authors wanted to answer questions such as: What difficulties can arise in joint planning of a blended course? How can teachers be supported in the planning and implementation? Secondly, the authors wanted to collect data on students' perspectives on joint international blended courses, including the motivation for their participation, the perceived potential of the course to facilitate international learning, perceptions of the course concept, etc. Lastly, the authors wanted to learn more about teachers' perspectives, e.g., what incentives and support do teachers want when creating and designing their joint international blended teaching? What challenges do they see in planning and implementing the course? etc. In order to find answers to these questions and to collect data, the authors gathered experience during the implementation of the joint international blended course and conducted two surveys of the participants at the end of the semester.

### 2.1. Implementation of the Joint Blended International Course

In order to attract participants to the experimentation, the OpenU project manager/E-Learning consultant from the Services for Teaching and Learning of the University Library of the FUB issued a call for proposals. The call was advertised at various information and networking events, as well as in training courses at the FUB. In the case of UCM, the call was e-mailed directly to all teaching staff by the Vice-Chancellor of Technologies and

Sustainability, who was also responsible for the overall coordination of OpenU at the UCM. In the end, two very motivated and engaged teachers from FUB and UCM were ready to offer a joint blended course in the field of "Sustainable Development" in the summer term 2022: Dr. Berthold Kuhn from FUB and Dr. Asun López-Varela from UCM. Dr Kuhn is a political scientist and works as a private lecturer at the "Otto-Suhr-Institut" and at the "Sustainability & Energy Unit" of FUB. Asun López-Varela is Assoc. Prof. at the Facultad Filología of Universidad Complutense de Madrid.

In July 2021, the requirements for a joint international blended course were discussed between FUB's OpenU E-Learning consultant and the Academic Coordinator for Sustainability (in) Teaching at the "Sustainability & Energy Unit". The number of course places available for FUB students was reduced to 15 instead of 30, in order not to increase the supervision effort due to the additional group of max. 15 students from UCM. It was agreed that access to the course should be as prerequisite-free as possible, as it primarily aimed at undergraduate students. Furthermore, as teaching and learning activities, the course had to include project-based collaboration and work in student groups.

More detailed planning for the course started in October 2021. The Academic Coordinator for Sustainability (in) Teaching and the OpenU E-Learning consultant together with both teachers, elaborated a framework for a possible teaching cooperation in the summer semester 2022, which included, i.a., the following aspects: How many online sessions and how many face-to-face meetings should be organised and when? How to deal with different ECTS requirements at the two universities? Which funds could be applied for in order to finance the physical short-term mobilities? Do the student groups match each other in terms of prior knowledge and fields of study?

The reasons for planning only one face-to-face phase at the FUB were, i.a., the only slightly overlapping lecture/semester times of FUB and UCM, especially in the summer semester: shared lecture period from 25 April to 10 May, including the exam period at UCM until 8 July 2022. In addition, the OpenU E-Learning consultant at FUB was only eligible to apply for funding for the internationalisation of teaching at FUB. In order to cover the travel costs of the FUB students to Madrid, it would have required either a longer period of presence at the UCM of at least 5 days in order to apply for funds for blended mobility, or the initiative of Madrid to apply for funds for the internationalisation of teaching at UCM. Eventually, funding was applied for from the Una Europa Early Career Host Programme to support joint teaching events at the FUB with partner universities of the Una Europa University Alliance [21]. These funds were intended to cover the travel costs to Berlin of a maximum of 15 UCM students and the teacher, Ms López-Varela. The administration of the reimbursement of travel expenses and the extensive communication with the UCM students were taken over by FUB's OpenU E-Learning consultant in cooperation with the Una Europa project coordinator.

Mr. Kuhn and Ms. López-Varela drafted the joint teaching concept, starting in February 2022. With the working title "Global Perspectives in Sustainability Transitions", the seminar sought to introduce the concept of sustainability and the 2030 United Nation Sustainable Development Goals from an interdisciplinary STEAM perspective. Thus, it included different approaches from various academic disciplines in working with sustainability concepts. As mentioned, sustainable development requires an integrated bottom-up approach that looks at the full spectrum of scales, networks, states, and shifts. The students gained knowledge on the UN framework which includes the 17 SDGs with specific sessions focusing on the following three pillars:

- Pillar 1. Social Progress and Health, including SDGs 1 to 7 as well as SDG11 and SDG16.
- Pillar 2. Economic growth and Circular bioeconomy, including SDGs 8 to 10 and SDG12.
- Pillar 3. Climate change, life underwater and life on land, including SDGs13 to 15.

They also learnt how sustainability is being approached by the general public of non-specialists, appearing, for example, in cultural representations that include art and literature.

The learning experience was centered on the students and their individual approach to sustainable development. Therefore, part of the online sessions was conducted in a flipped

classroom format, where students were to explore a topic or activity before class, ranging from UN reports, to videos, books, graphic narratives and cell-phone APPs about particular initiatives to help sustainability [22], and then share their findings with their classmates in an informal way during the online sessions, focusing on each of the pillars discussed.

The international joint blended course was designed on a semiotic basis: sustainability was explored moving upwards to the social level, in relation to initiatives taking place at the students' respective universities, their families, larger communities, regions and countries. Thus, alongside the theoretical introduction to the 17 UN Sustainable Development Goals (SDGs), this systemic approach involved careful attention to students' individual responses to their surrounding environments. It took into consideration the values they attached to things within their physical/perceptual/material realms, the forms in which these things acquired meanings, qualified and quantified, and were made sustainable. Finally, the social concerns that emerged when these aspects were also discussed and considered, both online and face-to-face in joint group interaction during the blended mobility exchange.

At the FUB, the course was offered as a "General Professional Skills Course" at the "Sustainability & Energy Unit". These courses are geared towards the acquisition of practice-related skills. They are part of all Bachelor's degree programmes at FUB: students usually have to acquire 30 ECTS of their undergraduate studies with courses from the General Professional Skills area. FUB students received 5 ECTS according to their learning and workload in the blended course. UCM students took the course complementary to their study programmes. In Complutense, undergraduate students can acquire up to 6 ECTS in extracurricular activities and classes. According to UCM regulations, since the course had a blended learning format, they were entitled to receive 2 ECTS. They did not have to take a final exam in the form of a poster presentation of a sustainability project at the end of the semester like FUB students.

While at the FUB undergraduate students from all disciplines were eligible to attend the course, at UCM, undergraduate students from the Faculty of Philosophy who had already taken a preliminary course with Ms López-Varela could participate. All students were admitted on a "first come, first served" basis and no official proof of English language proficiency was required at either university. A total of 14 FUB students from different degree programmes enrolled in the course. From UCM, 13 students participated, with the majority enrolled in the BA study programme "English Studies". During the course, the number of participants from UCM decreased to 11, due to exam coincidences at the end of the semester.

The course started with an online phase on 25 April 2022, taking place each Monday from 10 to 12 a.m., ending with a face-to-face phase at FUB between 8–9 July 2022. The course included 8 online sessions (total 12 h) and 2 face-to-face sessions (total 12 h), along with Q/A sessions (total 4 h). The online sessions were conducted using the FUB's video conferencing tool, Cisco Webex. UCM students did not need a separate FUB account to participate in the live online sessions but could participate via browser. Moreover, a blackboard course (LMS) was created for the course, which was mainly used for announcements by the teachers. To use Blackboard, UCM students needed their own FUB account. After confirming their accounts, students had to be manually added to the course in the LMS.

## 2.2. Student and Teacher Survey

The student survey consisted of 22 closed and 10 open question items on the topics "Seminar content", "Media didactics" (e-learning) and "Teaching skills". Under the section "Seminar content" the survey was to determine, i.a., the relevance of the international aspect of the course for the students' motivation and satisfaction. Moreover, the authors wanted to know whether the blended course, from the students' perspective, had the potential to provide international experiences and enable international learning in addition to acquiring subject-specific or methodological knowledge. International learning refers to the further development of intercultural and diversity competences; for instance, by getting to know different academic perspectives and approaches to the same subject, which

may vary from country to country. Furthermore, the survey aimed to determine whether the blended international teaching format could act as a steppingstone to physical mobility. Under the section "Media didactics" the survey should identify the students' perspective on the design of the blended course with questions about the blended course concept, the perceived balance of online and face-to-face as well as synchronous (i.e., live online sessions) and asynchronous (i.e., self-study, self-organisation) phases, and about the cooperation and communication with the students from the partner university during and outside the online sessions. Under the section "Teaching skills", the survey was intended to identify how students viewed the teaching concept and methods to determine what forms of facilitation and support the students would have liked from the teachers. All the individual questions of the survey and the students' answers can be found in the Supplementary Materials.

In order to achieve the highest possible response rate, the evaluation was scheduled in the last face-to-face meeting. The students received a PDF with a QR code and token (password) as well as a link to the online evaluation form. They were asked to take 10 min to answer the questionnaire and to consider the open questions as constructively as possible, i.e., clearly formulated, appreciative and with concrete alternative suggestions if something was critically assessed. The absent students received the evaluation by e-mail. They had until 31 July 2022 to complete it.

The quantitative part of the questionnaire was analysed and processed by the "Sustainability & Energy Unit" at FUB with the programme "Unizensus", the central evaluation software of FUB for teaching and course evaluations (see Supplementary Materials). The qualitative part was analysed and summarised by the authors. The student survey had a response rate of almost 100% (24 out of 25 students); the usual average response rate of students in FUB's elective courses is 30%.

The student data collected in the survey with Unizensus were not personal. It is, therefore, not possible to draw conclusions about specific survey participants from the collected data. All participants were informed that anonymity is assured and that their answers help to ensure or further develop the quality of teaching and study at FUB. Their participation was voluntary and there were no consequences for those who did not answer. In compliance with the European Regulation, specifically the General Data Protection Regulation (GDPR) applied to research in humanities and social sciences, the survey was conducted anonymously. This was guaranteed by various measures, e.g., the allocation of tokens (passwords or access to the online evaluation) or the anonymisation of handwritten comments in printed questionnaires.

To better identify the needs of the teachers regarding the planning and implementation of the course, they were asked to answer a qualitative questionnaire in the end of the semester with ten open questions on the topics of "planning and designing of the course", "blended course concept" and "student participation" via the ARS/evaluation tool Votingo. The focus was on perceived barriers to planning and teaching a joint international blended course, incentives that would facilitate the implementation of such courses, digital tools and support services that teachers would like to have for the implementation and a reflective engagement with their own course concept and teaching methods. The survey was analysed and summarised by the authors. It had a response rate of 100%.

## 3. Results and Discussion

### 3.1. Challenges in the Joint Planning of the Blended International Course

A major challenge in the planning was to find teachers who would be interested in a blended teaching cooperation in the Una Europa University Alliance and who would be willing to invest time in designing and planning a joint course. This may have been concerned with a certain 'digital fatigue' among the teachers who, after four semesters of mainly teaching online courses because of COVID-19, wanted to get back to campus and teach face-to-face again. The only slight overlap in lecture times between the FUB and other Una Europa universities, especially in the summer semester, also made it difficult to hold joint courses. Finally, the considerable extra effort required to plan a joint international

blended course should be mentioned as a possible reason for the lack of interested parties in the experimentation: Especially without prior contact and experience in tandem teaching, the design of a joint course or course requires more time to discuss learning objectives, create a syllabus, apply shared teaching methods, etc.

Ideally, the extra coordination and communication effort required for the joint planning of an international blended course should be balanced by less synchronous teaching/lecturing time and/or moderation time due to the second teacher. However, in this experimentation, more planning and coordination effort was necessary. The effort for supervising and examining students remained more or less the same as for regular courses, as each teacher was only responsible for his/her own group of students.

It is advisable to discuss the framework conditions and basic requirements for an international blended course with the course planner and the head of the respective department before planning a course to include course requirements, specifics of the subject area, strategic goals, etc. Strategically, it also makes sense to talk to the department heads about their interests and needs for digital, international teaching cooperation right from the start. The heads have better contact with the teachers, which can significantly increase the likelihood of feedback.

Planning a joint international blended course requires increased effort and the need for structured preparation (more than regular seminar planning). Early consultation with partners is, therefore, important: ideally, this occurs at least half a year before the planned start of the course, one year before is even better. A joint communication medium must be found for the planning: it is a good idea to create a shared document or wiki for the teaching concept, which the actors involved (teachers, teaching planning, e-learning, etc.) can access in order to always find information in one central place. Regular contact and communication between the teachers are indispensable for the planning process. If the teachers do not know each other from previous contexts, it may be advisable at the beginning of the cooperation to initiate and moderate the meetings from the project/teaching planning side. Factors in planning a joint blended course include:

- Financial framework: Identification and calculation of available funds and funds to be raised, application for funding
- Available time frame: Determine the start and end of the project, consider possible constraints through fixed dates such as the course of a semester, draw up a milestone plan with buffer zones and deadlines.
- Personnel capacities: Clarify responsibilities and division of tasks with the teaching partner; if applicable, include student assistants or research assistants
- Support from services for teaching and learning should be asked for at the beginning of the planning process and at important points during the project
- Technical aspects: Clarify the technical and E-Learning infrastructure at own and partner university and decide early on which systems you want to use

When designing a joint course, it is important to first discuss the content, objectives and learning goals together with the partner teacher. The content of one's own course should be expanded with the help of the other teacher's offerings and jointly developed beyond the original course content. In doing so, a certain openness to new orientations of one's own course should be demonstrated. The following questions can help in the beginning of the designing phase:

- Which (learning) tasks are suitable to support learners in acquiring competences through self-organised and cooperative forms of learning?
- Which forms of assignments and examination are best suited for working on these tasks?
- What information must be made available to the learners so that they can work on the tasks?
- Which information should they work out themselves and, if necessary, also make available to other course participants?
- Which learning tasks and information should be combined into learning units?

- What organisational and procedural plan, i.e., what learning scenario does this combination of learning tasks and forms of work suggest?

Next, the structure of the course must be determined together: The overall sequence of the course and the sequence of the learning units must be determined, as well as the alternation of asynchronous and synchronous learning phases: successful digital learning requires asynchronous collaboration on assignments as well as synchronous communication for exchange with teachers and peers. Synchronous meetings/presentations are useful at the beginning of the course to clarify the objectives and to reduce fear of contact between students. Expectations and requirements of students should be defined and aligned. The preferred social form should be international group work, i.e., self-organised, mixed international learning groups working cooperatively on selected learning assignments and seminar topics to enable international learning. Attention should be paid to a fair distribution of work between students, and a realistic amount of time should be considered for processing. Teachers should not rely on communication between students running by itself. There should always be occasions for communication. Jointly established rules for communication can prevent misunderstandings.

Teachers should also be comfortable with the technical resources needed to implement their learning scenario. Likewise, media that are widely used among students should be used whenever possible. Finally, the examinations must be determined: In joint international blended courses, the final examination can take place at the respective university.

### 3.2. Student Survey

3.2.1. "Seminar Content": Internationalisation

In addition to the professional introduction to the topic of sustainability development and SDGs, for most students, the exchange with an international group of students was the main motivation for their participation in the course. As expected, the UCM students additionally indicated the physical short-term mobility in Berlin as a major reason.

> I joined the curse [sic!] for two reasons: first, to continue participating in inter-university projects, and second, to have a chance to travel to Germany. My expectations were fully met since I was looking forward to a project that would emphasise teamwork (as well as individual participation and investigation) in relation to sustainability. Of course, the prospect of a trip was highly motivating and made me work even harder than if there had not been a "reward" of some sort.

(Supplementary Materials, p. 2)

However, many FUB students, though not travelling to Madrid, also gave the exchange with UCM students as a reason for their interest in the course. Comparative studies would be necessary to determine whether the physical short-term mobility at the partner university is the main reason for students to participate, or whether international exchange is the main focus, i.e., whether pure online courses with an international partner would have a similar appeal. Some students stated that they had taken part in the course to practise their English language skills. The international exchange with the students of the partner university was also mentioned as the most positive aspect of the seminar, closely followed by the simulation game, which was conducted on-site at the FUB campus.

In order to determine whether the course facilitated international learning, the students were asked whether they had been able to get to know different cultural or national perspectives on the topic of sustainability: 75% of students claimed that this statement fully applies and 20% said that it applies. Students should also indicate, how they had or had not benefited from the participation of an international group of students and from being taught by two international teachers. The main advantage of having a group of students from an international partner university in the course was said to be the different experiences, perspectives and backgrounds of the participants, which often led to interesting discussions.

> Treating with people from different countries is, I feel, essential to understand
> that we are not that different after all and that it simply requires more time to
> understand another point of view that may even challenge ours, but it always
> leads to growth.

(Supplementary Materials, p. 4)

As a benefit of an international teaching tandem, the students mentioned above all the different subject perspectives and academic cultures, in addition to the different teaching methods. No disadvantages were mentioned here. More comprehensive quantitative studies would, of course, be needed to determine the extent to which certain blended international courses could increase intercultural and diversity competences.

About half of the participating students had not participated in a physical exchange before the course. A total of 90% of all students claimed that the statement "After attending the course, I am more in favour of participating in a physical exchange program." fully applies, while the remaining 10% stated that this statement applies. In particular, students can imagine participating in a physical exchange programme at UCM or FUB: 85% said that this fully applies, 15% answered that this applies. This shows that blended international courses can contribute to students being more likely to consider a physical exchange.

3.2.2. "Media Didactics": Blended Learning

Most students liked the blended format, which they would prefer to purely online formats in the future. A total of 25% of the students would prefer hybrid formats for future international teaching collaborations, i.e., part of the teaching is on-site, part is digitally connected. As expected, the majority of students would have preferred a second face-to-face phase at the UCM, so that FUB students could also have a physical short-term mobility. This critique is particularly important because an imbalance in the physical mobility flow in a blended course could lead to a motivation gap among students, i.e., students who are allowed to travel to the partner university have an additional incentive for high participation and investment of work, which is lacking in the other student group. Some students would have liked to have a longer face-to-face phase at the FUB, because the participation of all students was higher and communication among each other as well as cooperation was better on-site. In addition, the students would have liked to have more time to get to know each other personally or to run more simulation games.

Most students liked the balance of synchronous (weekly online sessions) and asynchronous (i.e., self-study, self-organisation) phases during the online cooperation phase. There were no digital tools that were missing for the students, i.e., the learning management system of the FUB, Blackboard, and the video conferencing tool, Cisco Webex, were perceived as sufficient for teaching, learning and communication. However, it was critically mentioned that there had been no joint digital platform, so that the UCM students had to be manually entered into the LMS by the support of the FUB and students had to confirm manually.

The cooperation and communication with the students from the partner university during the online sessions was perceived as unbalanced: the majority of students stated that student participation in discussions varied greatly and that only a few students, especially from UCM, participated in joint discussions during the online sessions. At the same time, all students confirmed that the teachers encouraged the communication and collaboration between the students (Supplementary Materials). The reasons for the lower participation of FUB students, therefore, could lie in the aforementioned motivation gap due to the lack of the incentive of a physical short-term mobility in Madrid. The Academic Coordinator for Sustainability (in) Teaching of the "Sustainability & Energy Unit" saw the fact that the course at the FUB was offered as a "General Professional Skills Course" for undergraduate students as a further reason for the partly lower motivation of the German students. Commitment and participation would usually be lower in these courses compared to the regular study programme.

### 3.2.3. "Teaching Skills": Teaching Concept and Methods

Most students found that the learning objectives were clearly formulated at the beginning of the course (85%) and consistently implemented throughout the semester (80%). A total of 90% of the students claim that the teachers explained in an understandable way. Students liked that the seminar was interactive, participative and that students could work together in groups.

Several students expressed the wish for clearer instructions and work assignments, as well as more guidance from the teachers during the work phases. Some students would have liked more structured lectures and fewer student presentations. Others wished for more organisation and clearer expectations for the course and its requirements. In order to increase student participation, it was suggested, for instance, to have group work during the online sessions, with teachers facilitating and giving input during the work phases.

### 3.3. Teacher Survey

To determine the reasons for teachers' engagement and participation in the project, they were asked about their perceived potentials of international online and blended teaching:

1. Diverse online and face-to-face inputs from different disciplines and from different countries and regions would increase motivation and competencies of the students
2. Joint international online teaching would meet the requirements of the EU Commission Council Recommendation on building bridges for effective European higher education cooperation [23,24]
3. Joint international online teaching would meet the EU target that, by 2030, at least 45% of 25–34-year-olds obtain tertiary level attainment [25]
4. Universities would have a unique position at the crossroads of education, research, innovation, serving society and economy [26].
5. Joint international online would have the potential to strengthen the flow of knowledge also in research and innovation, enhancing transnational cooperation, creating a more inclusive and connected Higher Education, and helping build resilience and global competitiveness of European higher education system [27].
6. There would also be opportunities to maximise Europe's global influence when it comes to values, education, research, industry and societal impact, helping universities become lighthouses of the European way of life and reinforcing universities as drivers of the EU's global role and leadership.
7. Joint international online teaching could empower universities as actors of change in the twin green and digital transitions.

Moreover, the survey was to identify possible explanations for the few applications for international digital teaching collaborations. The following were cited as challenges in implementation of joint international online and blended teaching formats:

1. alignment of policy priorities and investments at EU, national, regional and institutional levels
2. the elimination of legal and administrative obstacles to international strategic institutional partnerships
3. structural and operational issues that include
   a. possible incompatible requirements
   b. diverse temporal frameworks
   c. different syllabuses that prevent the execution of programmes as well as the award of joint evaluations
   d. admission and enrolment criteria of students and lifelong learners
   e. defining the languages of instruction
   f. inclusion of flexible learning pathways
4. new instruments and legal frameworks for alliances
5. funding of universities is often insufficient to fulfil their growing societal mission. Additional funding is needed to help in fostering synergies [28].

6. significant disparities in digital skills across the EU must be overcome
7. quality assurance procedures, impact assessment European Quality Assurance and Recognition System needed.
8. similar infrastructures (e.g., in digital tools) should be in place

Teachers were also asked to identify possible incentives that they felt were needed at the departmental and/or university level to enable more joint digital, international teaching-learning formats to be offered in the future:

1. adequate compensation for extra time needed to engage in communication and exchanges with university administrations at different levels
2. recognise in their career assessment the time spent by academics in the development of new innovative pedagogies through transnational cooperation
3. adequate financial support
4. economically valorise a teacher's time in these activities, or else recognise them as part of their teaching workload
5. support online as well as face-to-face interactions, including short mobility exchanges.
6. ensure flexibility in funding programmes to allow for interdisciplinarity
7. administrative and tech support from higher education institutions
8. joint digital strategies and shared interoperable IT infrastructure
9. training and support services
10. seamless access to findable, accessible, interoperable and reusable (FAIR) data and other interoperable services
11. support capacity building for strong and effective leadership in implementing joint ventures

When asked what they think were the biggest challenges in the joint planning and designing of the course (before the semester), Mr. Kuhn indicated the different academic calendars between the international partners. In his opinion, this could be improved via communication and flexibility of students and administration. Another challenge was making teaching frameworks compatible in participant institutions and design common evaluation proposals for students. In both cases, higher management structures were involved, which made the process more complicated.

The teachers were also asked to indicate what they enjoyed about teaching together with their international partner and what challenges they encountered during the semester. Ms. López-Varela expressed, that it was interesting to learn how things occurred in the partner university, and that the major challenge was concerned with the timing of activities in the course. Mr. Kuhn stated that the different subject perspective of the partner teacher was very enriching, but that communication and administrative issues had taken up a lot of time.

Both teachers stated that they did not need any further digital tools for the joint curriculum/course planning or for the online teaching activities, implying that the available e-learning infrastructure of the FUB, specifically the LMS Blackboard and the videoconferencing tool Webex, were sufficient for their purposes. Mr. Kuhn though would have preferred the use of a shared digital/e-learning infrastructure in order to avoid an extra effort for getting a group of students used to the digital infrastructure of the partner university.

According to the teachers, there was also no particular need for support from the different support services at the university regarding the planning and the teaching of the course. However, it should be noted that the Academic Coordinator for Sustainability (in) Teaching at the "Sustainability & Energy Unit" as well as the OpenU E-Learning consultant of the FUB intensively supported the teachers in planning the course, especially in working out the framework conditions of a joint international course. In addition, the Una Europa project coordinator of the FUB supported the management of the physical short-term mobilities of the UCM students.

The teachers did not use specific teaching methods, concepts and/or formats to enable international learning. Cooperation among students was principally facilitated through mixed group work, whereby the students were largely given a free hand in group composition and working together. Ms. López-Varela stated that she would have used

inverted/flipped classroom methods. With more time, she would have liked to introduce problem-solving activities using Design Thinking methodologies, for she thinks this is the future of higher education.

The teachers were asked how they rated the participation and motivation of students in the course and, if applicable, what reasons they saw for lower participation/motivation of specific students: The participation and motivation of the Spanish students was in general perceived to be higher. A possible reason for this mentioned was the incentive for the Spanish students to be able to travel to Berlin as part of the attendance phase of the course, with their travel costs being covered. In addition, UCM students were made aware that the physical short-term mobility in Berlin would require their active participation during the online sessions. Therefore, both teachers recommended having two attendance phases for future blended teaching cooperation: one at the beginning of the cooperation at one university, the second at the end of the cooperation at the partner university. Students would be more motivated to collaborate and communicate when they have met in presence. In Mr Kuhn's opinion, ECTS and certificates would provide the best incentives for students to actively participate in a course.

## 4. Conclusions

The following is a summary of some of the main findings and conclusions from the experimentation between the FUB and the UCM.

### 4.1. Opportunities of Joint International Blended Courses

Blended mobility formats have the potential to significantly increase student mobility at universities and other HEIs by providing a higher number of students with the opportunity to have an international experience without the need to participate in long physical exchange programmes. They combine the advantages of virtual and physical mobility: the digital cooperation phase makes BM formats to be scheduled more flexibly into students' study programmes and daily lives. Blended mobility is, therefore, more inclusive, offering international experiences especially to those who are limited in their ability to participate in longer physical mobility due to family, financial or other reasons. At the same time, the included face-to-face phase makes blended mobility formats more attractive for students than pure online courses.

Our student survey showed that students choose certain courses not only because of the course content but also because of the participation of an international student group and teacher from a partner university. Internationalisation of teaching and learning with BM formats can, therefore, make study programmes more attractive for students. International blended courses can enable international learning, if teachers encourage communication and collaboration between student groups through the course structure, meaningful assignments, provision of appropriate digital tools and motivational encouragement. Blended mobility can also become a steppingstone for students to physical exchange programmes, as it provides a low-threshold insight into the teaching programme, teaching methods, student life, etc., at an international partner university.

Students would like to attend blended teaching-learning formats that are as interactive and participative as possible, with clearly formulated assignments and requirements and a high proportion of group work, to promote the participation of all students in joint discussions and collaborations. Accordingly, students should be able to become as active as possible during the online sessions, be it through discussions, presentations, group work or similar. Cooperation between the international student groups should be promoted through meaningful tasks and methods, communicated as clearly as possible, e.g., working together on long-term tasks in mixed groups.

Joint international blended courses could be integrated into the existing curriculum of a study programme, if blended learning is recognized in the respective study and teaching regulations, and the overall students' work and learning efforts remain the same compared to the usual face-to-face course. When implementing a joint international course, both

groups of students should receive the same number of ECTS credits for taking the course within the framework of their respective study programmes and regulations. Otherwise, students would have to work different hours, which could make it difficult to work together and cooperate. Teachers would have to be prepared to open their curricula and teaching methods, and to design a joint course curriculum in exchange with the partner teacher. Students would not necessarily have to be enrolled at the partner university as part of their mobility, since they would not have to take (e-)examinations at the partner university but could be examined at the end of a semester by their own teacher at their own university as usual.

*4.2. Challenges in the Implementation*

A major challenge in the implementation of international blended courses is to find teachers who would be interested in a blended teaching cooperation and who would be willing to invest time in designing and planning a joint course. Experience shows that it is easiest if the teachers already know each other from research collaboration or other academic contexts. In this case, the motivation for a teaching cooperation is usually high enough so that no further incentives are needed. In the case of new cooperation, however, incentives are necessary for the teachers, be it intensive support in the planning and implementation of the course by the support services for teaching and learning, and for the organisation of the physical short-term mobilities also by the International Office, be it crediting of the extra effort to the teaching load or be it funding. One approach to recruit interested teachers for international blended teaching collaborations is to first seek dialogue with the department's heads and management, to convince them of the benefits of international blended teaching and learning scenarios (international experience for *all* students), include their needs and perspectives on the internationalisation of teaching, and eventually to agree on specific targets for offering blended courses. Teaching staff could then be invited to express their interest in offering international blended courses.

Another challenge lies in convincing the relevant stakeholders at the departments and faculties to reduce the course places in a joint blended course that are available to their own students (compared to a regular face-to-face course). For example, a course that usually has 30 places for FUB students will only have 15 places in a joint international format, as the others are taken by students from the partner university. Theoretically, it would be possible to increase the number of total course places, but despite a second participating teacher from the partner university, this would significantly increase the supervision effort. Even if the teachers share the moderation of the sessions, the supervision of group work or the evaluation of the students' performance, certain teaching-learning activities, such as joint discussions or student presentations, are not feasible with too many participating students.

When designing international blended learning courses, it is important to conduct a face-to-face phase at each participating university so that no student group feels disadvantaged, and the motivation and participation of all participants is equally high. Ideally, an attendance phase is carried out at the beginning of the cooperation so that the students can get to know each other personally and, thus, create a good basis and high motivation for the collaboration during the online phase. A presence phase at the end of the course at the other partner university could then be used for a discussion of the joint work and for final presentations. If there is only a slight overlap in the lecture times of the universities, it makes sense to use these for the online cooperation and to schedule the attendance phase as a kind of summer school during the lecture-free period. The incentive of financed short-term physical mobility increases the likelihood that students will also attend courses outside the lecture period. Of course, this in turn requires flexibility on the part of the students as well as the teaching staff, administration and teaching support.

Travels to the partner universities should ideally occur via train/sustainable travel options within Europe. Therefore, longer attendance phases should be organised, so that students and teachers can travel by sustainable means of transport and the travel time and travel costs are at the same time in a reasonable proportion to the length of stay. We

recommend the consideration of climate concerns in the implementation of joint blended courses at Una Europa universities and advocate for the inclusion of sustainability criteria in the funding criteria of Una Europa funding.

### 4.3. Requirements for the Joint Digital Platform (BLOOM)

The establishment of digital and cross-location infrastructures is in most cases the first step towards the development of joint digital teaching-learning offerings cf. [29] (p. 148). In order to address the important challenges of internationalisation and sustainable development, the OpenU consortium is creating a European digital hub, called BLOOM, that is intended to provide a joint digital infrastructure, which should support the design and implementation of joint teaching, cooperation and mobility formats in Una Europa and the European Higher Education Area.

Future international blended international courses could be hosted and implemented via BLOOM. One big advantage would be that participating students could use the account from their home university to access the offered courses via eduGAIN, without the need of having received a second university account at the host university for gaining access to all digital platforms and resources. The providers of the platform would then have to decide which attributes of the students are needed and must be transferred from the home university of the student. This procedure would require at least a suitable master data record to be created at the host institution, to which, e.g., course bookings or examination results could be linked.

The implementation of a course catalogue on the hub with all available international blended courses for Una Europa students would be useful, as international BM formats cannot yet be flagged in the Campus Management/Student Life Cyle Management Systems of the FUB or UCM.

An eTwinning tool should be integrated on the joint platform so that interested teachers can submit search queries for potential cooperation partners and teaching tandems can be found more efficiently. Until now, teachers either had to already know each other or the project coordinators had to undertake a lot of communication and advertising at the different partner universities to find potential partner teachers.

A sustainable support concept and corresponding personnel resources are needed to answer the requests of teachers and students regarding the platform and to support them in the design/concept of digital teaching and learning scenarios as well as implementation of and participation in the blended courses.

It would be preferable if the blended courses could be offered via the HEI specific e-learning infrastructure. Teachers prefer to use the e-learning systems and digital tools they are accustomed to and often do not have the capacity to familiarise themselves with new LMS, video conferencing software, etc. This is strongly regarded as an obstacle to teacher motivation in offering BM concepts in their own teaching. Therefore, BLOOM should be able to integrate the HEI specific e-learning infrastructures.

### 4.4. Policy Implications

In general, motivated early adopters and bottom-up drivers among teachers, departments, and service center staff are essential for the introduction of blended mobility formats at HEIs. For a long-term implementation of BM formats, commitment and investment from decision-makers at the middle (department head, dean of study) and higher institutional levels (university board, management of the International Offices or centers for international cooperation) are equally important cf. [30] (p. 11).

On the departmental level of a university/HEI, target agreements could be negotiated between the department heads and the university management to allow for a certain quota of international blended and/or online courses. To enable teachers to offer blended courses, it is advisable to adapt the study regulations accordingly, i.e., inclusion of blended teaching in "teaching and learning format" regulations and include the blended mobility format in the "study abroad regulations".

On the university/HEI level, the implementation of BM formats could be included with further details in the HEI's internationalisation strategy [1] (p. 13).

Furthermore, the university management, together with HEI policy on the federal state level, could decide on a regulation that integrates international blended teaching hours/preparation workload to "count" towards the fulfilment of teaching obligations under certain conditions.

On the federal state level in Germany, BM formats should be included in higher education contracts with the federal states (in Berlin 2023–2026) under the section "Internationalisation of Berlin's higher education institutions (internationalisation of teaching)" [31] (Berlin uses the instrument of higher education contracts to guarantee Berlin's HEIs planning and continuity in funding. It makes the amount of its subsidy dependent on the fulfilment of specific targets that are negotiated with the universities).

On the national level, government initiatives should continue to offer additional funding programmes for HEIs to support European University Networks and support programmes to enable the use of digital learning technologies and methods. The German Rectors' Conference (HRK) could negotiate and formulate jointly accepted standards and procedures for implementing BM formats at HEIs.

On the EU level, the further development of a joint European platform for digital education (e.g., BLOOM) should, of course, be promoted within the framework of the Digital Education Action Plan. Furthermore, the Digital Education Action Plan or the ERASMUS Charta could promote BM formats as inclusive, equivalent to physical exchange in order to motivate wider student participation.

A particularly important measure at EU level would be to further develop the Blended Intensive Programmes (BIP). BIPs are a great innovation in the current Erasmus+ programme generation 2021–2027 to foster the development of BM formats in teaching. They are intended to encourage the development of short, intensive and joint curricula and activities to provide students and university staff with the opportunity to participate in a short physical group mobility (5–30 days) combined with a digital phase. A minimum of 15 learners (students and/or staff) must participate. The funded students should be able to achieve at least 3 ECTS and at least 3 ECHE universities (Erasmus Charter for Higher Education) from 3 different programme countries are required for the conceptual development [32] (p. 9).

One main concern of the authors is that no requirements are formulated for the design of the digital cooperation phase in BIPs. Only a "virtual component description" is required, and innovative teaching methods are to be used, e.g., research-based learning or challenge-based approaches [32] (p. 7), though the latter could also only be used in the presence phase. However, this could lead to the online phase remaining under-complex, e.g., by only providing materials such as PDFs for reading preparation in an LMS or for short "get to know each other" meetings. This can be all the more significant because the relationship between online and face-to-face phases is not clarified either, e.g., whether they have to be related to each other in a didactically meaningful way. For the area of individual blended mobilities, at least some examples of the design of the virtual phases are given: "The objective [of blended mobility] is to facilitate collaborative online learning exchange and teamwork. For example, the virtual component can bring learners from different countries and study fields together online to follow online courses or work collectively and simultaneously on assignments that are recognised as part of their degree." [32] (p. 4) In the author's opinion, the requirements for BIP applications also need to be supplemented with criteria on how to design an effective virtual phase so that the blended mobility scenario can enable international learning or teaching skills, e.g., students have to watch learning videos and solve matching tasks, or students have to work on long-term tasks in mixed groups. The concept development of the BIP consequently would require not only "the cooperation between the international office and the faculties/departments, where a person will be assigned as blended intensive programme coordinator" [32] (p. 11), but also the e-learning support services of the HEI. Another need for change is the number of face-to-face phases:

within the framework of a BIP, only one attendance phase at one of the participating HEIs is allowed, but as our experimentation shows, both students and teachers expect short-term physical mobility to take place at all universities involved in a blended course or BIP. If there is only one attendance phase at a university, this can lead to motivational gaps among students who feel unfairly treated.

Finally, the implementation of a BIP as "enhancement of an existing [study] programme" [32] (p. 7) seems difficult to the authors. Since most study programmes at ordinary universities are predominantly face-to-face, the short-term physical mobility of at least five days taking place in the context of a BIP leads at least to the obligation for students to stay away from other courses, which is not ideal. Therefore, the implementation of BIPs as supplement to the regular study programme in non-lecture periods or as part of a joint degree programme seems more feasible. Shorter mobilities than 5 days, as in our project, are difficult to imagine for reasons of sustainability, since for such short-term mobilities it usually only makes sense to travel by air.

**Supplementary Materials:** The following supporting information can be downloaded at: https://www.mdpi.com/article/10.3390/educsci12110810/s1. Data from the student survey.

**Author Contributions:** Conceptualization, R.P. and A.L.-V.; methodology, R.P.; investigation, R.P.; writing—original draft preparation, R.P. and A.L.-V.; writing—review and editing, R.P.; project administration, R.P. All authors have read and agreed to the published version of the manuscript.

**Funding:** The APC was funded by Erasmus+ KA3 project "Online Pedagogical Resources for European Universities", Ref. 606692-EPP-I-2018-2-FR-EPPKA3-PI-POLICY (Acronym OpenU).

**Institutional Review Board Statement:** Ethical review and approval were waived for this study since the surveys were conducted anonymously, so no personal data were collected in the surveys. The research complies with the general evaluation guidelines of the FUB (https://www.fu-berlin.de/en/sites/qm/steuerung/_inhaltselemente/evaluationsrichtlinie.html) and the ethics of our institutions.

**Informed Consent Statement:** Informed consent was orally obtained from all subjects involved in the study.

**Data Availability Statement:** Not applicable.

**Acknowledgments:** The authors would like to thank Nora Große, the Academic Coordinator for Sustainability (in) Teaching at the "Sustainability & Energy Unit" of FUB, and Simon Rienäcker, the Una Europa Project Coordinator at FUB, for their intensive support of our project.

**Conflicts of Interest:** López-Varela was involved as a teacher in the experimentation, but not in the collection, analyses, or interpretation of data.

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
