# Peer review of "Internationalisation of Teaching and Learning through Blended Mobility: Potentials of Joint International Blended Courses and Challenges in Their Implementation"

_education, doi:10.3390/educsci12110810_

Round 1

Reviewer 1 Report

Well written, interesting and valuable paper on some results from EC projects, following intentions of sustainable development (UN/EU) also in course design and content.

A report from valuable project experiences often becomes more narrative than exploratory, since the interventions are made in a project primarily, and then reported, interventions /experiments are not initiated only with the purpose of publishing. However, I think this kind of articles emanating from projects is an important part of the research literature.

I lack more literature review and possibly more in-depth theory about “blended”, blended mobility, ICT integration into existing educational frameworks, the time and pacing & synchronous/asynchronous dimension of blended teaching/learning, etc. Authors as C Dziuban, P Moskal, C Graham, M Power, ND Vaughan, M Bower, etc may be helpful. What constitutes literature review and references now is mostly policy documents from UN and EC and project reports – which is as far as I can see very well done.

I find it peculiar and not mainstream, but perhaps acceptable ( - a question for the journal editor?) to name university and project officials in the text. That would normally be acceptable only if they also are authors referenced in the manuscript, or if they are very generally known authorities or very public persons, as heads of states, possibly. Otherwise their names should be taken away and their function implied instead. To write out people’s names also has GDPR implications (have they given their written permission?)

It would have been good with some point of reference of comparison to a normal student population. The students involved here seem to be quite active individuals that not fear extra work for being part of something new and possibly exciting. They have actively chosen this. How do these students relate to a normal student population? How much are this kind of courses possible to implement as normal courses?

The conclusions are well made, but perhaps some reflections could be added on if this kind of “blended  mobility” courses can in the longer perspective become a part of “the new normal” of higher education?

Reviewer 2 Report

The topic addressed in this work is very interesting.

References used are current and relevant.

It is necessary to describe participants and instruments in depth in the Method section and in Conclusions to specifically state the limitations and prospective.

the article is really of quality and I do not consider that further modifications are necessary.
The title fits the content.
The abstract follows the appropriate structure and has a timely length. The theoretical framework is built on current and relevant references, although it is always possible to make an effort to increase the number of references in recent years and their internationalization.
The sample is considerable.
The results are properly presented and the references have been prepared following the regulations.
It is also possible to go deeper into the limitations and prospective.

 In fact, compared to other works received, this one is of better quality.

Author Response

Thank you for your review and feedback. We will expand the methods section to include the aspects you suggested.

Reviewer 3 Report

In “Internationalisation of teaching and learning through blended mobility: Potentials of joint international blended courses and challenges in their implementation,” the authors present the implementation of blended mobility and survey results based on a course that students jointly took from two universities: Freie Universität Berlin and Uni-13 versidad Complutense de Madrid. Overall, the paper provides a helpful case study on how blended mobility can take place between two international institutes of higher education with notes on challenges and key planning steps.

Unfortunately, it is hard to follow the main goals and points of the paper as currently written. The authors seem to focus more on the presentation of the case but also include survey results as the main results, but the research questions and design of those surveys are not mentioned in the earlier sections of the paper. The methods section should be rewritten to focus on the main points in the development of the course, the structure of the course, and experience instead of the many details on how the collaboration started and who was involved or specific dates events happened. Significantly, any mention of the survey data collection is missing from the methods section. Some of the methods are given in 3.2 or 3.3, but the appropriate information should be moved to methods or added new to the methods section. Related to section 3.2, are there references to cite that your survey questions or strategy were based on? Did the survey and implementation go through IRB approval for research involving human subjects? Why are all the figures of the survey results supplemental instead of summarized in one table or figure in the paper? For other sections, others would be able to take notice of the key points easier if the information was organized into a table or figure such as Factors in planning a joint blended course or challenges of the project.

Round 2

Reviewer 3 Report

Thanks for your replies and revision in response to the review of "Internationalisation of teaching and learning through blended mobility: Potentials of joint international blended courses and challenges in their implementation." The revisions address the major concern of not including the survey information in the methods (as another reviewer also mentioned) and the research questions not being clear from the start to include the survey. Most of the items have been addressed. I still think it is unnecessary to mention specific names of individuals (also mentioned by another reviewer) in the methods section. I'm not sure how it helps others understand how to replicate or model the experience for a research journal. That would be the major edit that I would suggest at this point - to remove them. Overall, the contribution is thorough about the implementation and survey responses and should be useful to others considering how to execute a similar collaboration and course.  

Author Response

Thank you very much for your feedback. We are not mentioning specific names of individuals anymore.

The items "Supplementary Material", "Author contributions", "Funding", "Institutional Review Board Statement", "Informed Consent Statement", "Data Availability Statement", "Acknowledgments",
"Conflicts of Interest" at the end of the article were deleted after the first round of reviews. We had therefore added them back to the manuscript as "suggested changes". In the current manuscript version, almost all items except "Supplementary Material" and "Author contributions" are missing again. Is there a reason for this? I have added the points again.